# Comparison of Elixhauser and Charlson Methods for Discriminative Performance in Mortality Risk in Patients with Schizophrenic Disorders

**DOI:** 10.3390/ijerph17072450

**Published:** 2020-04-03

**Authors:** Kuan-Yi Tsai, Kuan-Ying Hsieh, Shu-Yu Ou, Frank Huang-Chih Chou, Yu-Mei Chou

**Affiliations:** 1Department of Community Psychiatry, Kaohsiung Municipal Kai-Syuan Psychiatric Hospital, Kaohsiung 80276, Taiwan; kuanyi1970@yahoo.com.tw; 2Department of Child and Adolescent Psychiatry, Kaohsiung Municipal Kai-Syuan Psychiatric Hospital, Kaohsiung 80276, Taiwan; isanrra@gmail.com; 3Graduate Institute of Medicine, College of Medicine, Kaohsiung Medical University, Kaohsiung 80708, Taiwan; 4Department of Anesthesiology, Kaohsiung Veterans General Hospital, Kaohsiung 81362, Taiwan; clairerurup@yahoo.com.tw; 5Kaohsiung Municipal Kai-Syuan Psychiatric Hospital, Kaohsiung 80276, Taiwan; f50911.tw@yahoo.com.tw; 6Department of Physical Therapy, Shu-Zen Junior College of Medicine and Management, Kaohsiung 82144, Taiwan

**Keywords:** schizophrenia, National Health Insurance Research Database (NHIRD), Charlson Comorbidity Index score, Elixhauser comorbidity index score, mortality

## Abstract

Although Charlson Comorbidity Index scores (CCIS) and Elixhauser comorbidity index scores (ECIS) have been used to assess comorbidity in patients with schizophrenia, only CCIS, not ECIS, have been used to predict mortality in this population. This nationwide retrospective study investigated discriminative performance of mortality of these two scales in patients with schizophrenia. Exploiting Taiwan’s National Health Insurance Research Database (NHRID), we identified patients diagnosed with schizophrenia discharged from hospitals between Jan 1, 1996 and Dec 31, 2007. They were followed up for subsequent death. Comorbidities presented one year prior to hospital admissions were identified and adapted to the CCIS and ECIS. Discriminatory ability was evaluated using the adjusted hazard ratio and Akaike information criterion (AIC) and Harrell’s C-statistic. We identified 58,771 discharged patients with schizophrenic disorders and followed them for a mean of 10.4 years, 16.6% of whom had died. Both ECIS and CCIS were significantly associated with mortality, but ECIS had superior discriminatory ability by a lower AIC and higher Harrell’s C-statistic (201231 vs. 201400; 0.856 vs. 0.854, respectively). ECIS had better discriminative performance in mortality risk than CCIS in patients with schizophrenic disorders. Its use may be encouraged for risk adjustment in this population.

## 1. Introduction

The Charlson Comorbidity Index Score (CCIS), calculated based on 19 comorbid conditions, was developed by Charlson et al. in 1987 by reviewing hospital charts and assessing their relevance to one-year mortality [1]. The weight assigned to each comorbid condition ranged from one to six and the individual score for each of these conditions contributed to the final score. The CCIS has been used extensively to evaluate the impact of comorbid conditions in a variety of conditions [2,3,4]. CCIS has been modified in several ways including by a reduction in the number of conditions considered from 19 to 17 [5] and the use of coding by specific International Classification of Diseases, Clinical Modifications (ICD-CM) diagnosis codes [6]. CCIS has been used previously to assess medical comorbidity in the schizophrenic population [7].

The Elixhauser Comorbidity Index Score (ECIS) is another popular comorbidity assessment method introduced by Elixhauser et al. in 1988. It incorporates thirty comorbid conditions and has been validated in acute-care inpatient hospital settings using administrative data [8]. A growing number of studies have suggested that the Elixhauser method is a better comorbidity risk-adjustment model than CCIS [4,9].

More than half of patients with schizophrenia have one or more medical comorbidities [10]. Their medical comorbidities may be related to several factors such as the cognitive and behavioral impairments associated with schizophrenia itself or the adverse effects of the antipsychotics prescribed to them [11,12]. Patients with schizophrenia have high rates of diabetic mellitus [13,14], hyperlipidemia, obesity [15], chronic obstructive pulmonary disease, and hypertension [16]. Furthermore, patients with schizophrenia have shorter lifespans than the general population [17,18,19]. In a meta-analysis, increased overall mortality rates were noted in patients with schizophrenia and other nonaffective psychotic disorders (standardized mortality ratio (SMR) = 3.09, 95% CI 2.9–3.3) [20]. Increased mortality rates for suicide were 12 to more than 20 times more expected and the increased standardized mortality ratio from natural causes was at least two times more expected in patients with schizophrenia [21]. Therefore, schizophrenia, itself, has been associated with a substantial chronic medical burden for multiple comorbidities impacting mortality. Moreover, older age, female gender, depression, and cognitive impairment have also been associated with increases in physical comorbidities in this population [22].

CCIS has been validated in its prediction of mortality in patients with schizophrenia [23]. However, it is not known if ESIC can be used to predict mortality in this population because it has only been used to access morbidity in them [24]. Both CCIS and ESIC have been used to access comorbidity in individual patients in health services research [25]. However, no previous study has compared the two tool’s ability to predict mortality in patients with schizophrenia. In this retrospective cohort study, we tapped Taiwan’s National Health Insurance Research Database (NHIRD) to identify incidences of comorbid conditions and evaluate their impact on the mortality of patients with schizophrenic disorders. We then used these data to compare the discriminative performance of CCIS and ECIS in mortality risk in this population.

## 2. Materials and Methods

### 2.1. Study Design and Data Sources

All data, including schizophrenic disorders, numerous comorbidity conditions, and death were collected from the National Health Insurance Research Database (NHIRD). Taiwan’s National Health Insurance program is a compulsory health insurance program established in 1995. It covers the outpatient, inpatient, emergency, and dental care of approximately 99% of Taiwan’s 23 million residents. Taiwan’s National Health Research Institute maintains a large representative population-based claims database with data provided from the Taiwan’s Bureau of National Health Insurance. This database is known as the National Health Insurance Research Database (NHIRD). This database contains comprehensive information on enrollees in the program; diagnoses listed using the International Classification of Diseases, Ninth Revision, Clinical Modification (ICD-9-CM); medical services rendered; and medications prescribed. Its data is released for research purposes and confidentiality of the enrollees is maintained as directed by the National Health Insurance Bureau. The data used in this study came from a subset of the NHIRD known as the Longitudinal Health Insurance Database, which consists of systematically collected and randomly sampled data obtained from the NHIRD during the period from 1996 to 2012. There are no significant differences in distribution of gender, age, or income level between the patients in this longitudinal Health Insurance Database and patients in the NHIRD.

### 2.2. Ethical Statement

This study was approved by the Institutional Review Board of Kaohsiung Veterans General Hospital, Taiwan (Ethical code: VGHKS15-CT12-01). Review board requirements for written informed consent were waived because all personal identifying information had been removed from the dataset prior to the analysis.

### 2.3. Study Population

In this retrospective cohort study, we identified all patients who were newly discharged with schizophrenia in the Longitudinal Health Insurance Database between January 1, 1996 and December 31, 2007. Patients with schizophrenic disorders, including schizophrenia and schizoaffective disorders were identified by the ICD-9-CM code 295 listed on their claims submissions. We excluded patients who were younger than 18 years old, patients who had not been discharged yet, or those who had some data missing. After exclusion, we were left with of 58,771 discharged patients with schizophrenia between 1996 and 2007 to include in our analyses. If a patient had more than one admission during this period, only the first admission data were entered in our analyses. All patients were followed until death or December 31, 2012—the end of the study period.

We also collected data on parameters such as age, gender, geographic region, socioeconomic status (SES), and designated level of the medical facility. The recoding of SES in the insurance database is based on average income associated with four different job categories: EC 1 (civil servants, full-time, or regularly paid personnel with a government affiliation); EC 2 (employees of privately owned institutions); EC 3 (self-employed individuals, other employees, and members of the farmers’ or fishermen’s association); and EC 4 (veterans, members of low-income families, and substitute service draftees). We used these EC categories to classify patients into three SES subgroups: High SES (EC 1 and EC 2), Moderate SES (EC 3), and Low SES (EC 4).

### 2.4. Comorbidity Identification

Tapping administrative and ambulatory claims records, we were able to identify and record comorbidities present one year prior to the admission for schizophrenia. We used diagnoses codes (ICD-9-CM codes) from administrative and ambulatory claims records to calculate the Charlson and Elixhauser index by using Cheng et al.’s (2016) version [9]. CCIS has 17 disease categories and ECIS has 30 categories. We adjusted for these differences following the previous suggestions [9]. We calculated the weighted Charlson comorbidity score and Elixhauser comorbidity score as previously described [1,26].

### 2.5. Statistical Analysis

Cohort entry time started on January 1, 1996 and end-of-study was December 31, 2012. Seventeen-year overall mortality was calculated. Death was recorded as an event, and defined by suspension from the National Health Insurance. The prevalence of comorbidities was calculated and the outcome was measured by post-discharge mortality. Patient characteristics were analyzed descriptively and group differences were tested using independent t-tests and Chi-squared tests. Cox proportional hazards regression was used to identify the risk factors associated with death in the whole sample. The Akaike information criterion (AIC) and Harrell C (C-statistic) were used to assess predictive performance and evaluate discrimination against base model parameters (age, gender, SES, geographic area, and teaching level of hospital). The AIC was calculated as AIC = 2k − 2 ln(L), where k is the estimated parameter in the model and L is the maximum likelihood [27]. The smaller the AIC, the better the predictive ability of the model. The Harrell C-statistic measures how well the model can discriminate between observations, with possible values of 0.5 (no predictive ability), 0.7 to 0.8 (acceptable), 0.8 to 0.9 (excellent), 0.9 to 1.0 (outstanding), and 1 (perfect discrimination) [28].

SAS statistical software for Windows, Version 9.3 (SAS Institute, Cary, NC, USA) was used for data extraction, computation, linkage, processing, and sampling. All other statistical operations were performed using SPSS statistical software for Windows, Version 17 (IBM, Armonk, NY, USA). A *p*-value < 0.05 was considered significant.

## 3. Results

In total, we identified 58,771 discharged patients with schizophrenic disorders in NHRID from 1996 to 2007. Mean follow-up time was 10.4 years, and 9728 of the patients (16.6%) died. As can been seen in Table 1, the mean age of these patients was 37 years old, they were predominately male, and 43.7% were living in northern Taiwan. Almost half (49%) were classified into a low socioeconomic status, and almost fifty-two percent (51.9%) were discharged from regional hospitals. The most likely patients to die were those who were male, those who were older, those living in eastern Taiwan, those belonging to low SES, or those discharged from district hospitals. In addition, we found a significant association between higher CCIS and ECIS and increased risk of mortality (HR = 1.51, 95% CI = 1.49–1.54 and HR = 1.09, 95% CI = 1.09–1.10, respectively).

Table 2 and Table 3 show the distribution of comorbidities and mortality rates for the patients across Charlson and Elixhauser comorbidity scores. We could not calculate the comorbidity rate and mortality rate of acquired immunodeficiency syndrome (AIDS) or human immunodeficiency viruses (HIV) because there was no code for them in the NHIRD. A total of 8287 patients (14.1%) had comorbidities included in the CCIS. The most common CCIS-listed morbidities in all patients were peptic ulcer disease (4.2%), followed by diabetes mellitus without end-organ damage (3.3%), chronic pulmonary disease (3.2%), and mild liver disease (2.3%). However, in those that died, the most common CCIS-listed morbidities were, metastatic solid tumor (74.2%), followed by moderate liver disease (72.0%), myocardial infarction (64.6%), and congestive heart failure (62.6%). A total of 13,589 patients (23.1%) had ECIS non-psychosis (non-schizophrenia) comorbidities. In all patients, the most common ECIS-listed morbidities were depression (6.0%), followed by hypertension (4.9%), diabetes, uncomplicated (2.9%), and liver disease (2.6%). In those that died, the most common ESIC-listed morbidities were metastatic cancer (76.7%), followed by pulmonary circulation disorders (70%), congestive heart failure (63.6%), and renal failure (63.0%).

Table 4 and Table 5 summarize the results of our Cox proportional hazards regression analysis conducted to calculate the hazards ratio (HR) for mortality by CCIS and ECIC. In the univariate hazard ratio (aHR) analysis of CCIS-listed comorbidities, all items were distinguished as risk factors for mortality (HR = 1.47–10.08, *p* < 0.001) (Table 4). After adjustment for the patient’s age, gender, SES, geographic region, and teaching level of hospital, all the comorbidities remained associated with significant increases in mortality risk (adjusted HR = 1.21–6.31, *p* < 0.001). We found no factor to significantly lower the risk of death. In our univariate analysis of the association of ECIS-listed comorbidities and mortality, we found six items (hypothyroidism, lymphoma, obesity, blood loss anemia, drug abuse, and depression) that had no significant association with mortality (Table 5). In our multivariable aHR analysis of ECIS-listed comorbidities and mortality, however, almost every comorbidity was a risk factor except peripheral vascular disorders (aHR 1.19, 95% confidence interval (CI) 0.78–1.79), hypothyroidism (aHR 1.08, 95% CI 0.73–1.58), peptic ulcer excluding bleeding (aHR 1.24, 95% CI 0.96–1.59), lymphoma (aHR 1.21, 95% CI 0.39–3.75), rheumatoid arthritis/collagen vascular diseases (aHR 1.10, 95% CI 0.83–1.47), obesity (aHR 1.44, 95% CI 0.82–2.54), and blood loss anemia (aHR 1.51, 95% CI 0.89–2.55). No ESIC-listed comorbidity protected the subjects against mortality (Table 5).

As can be seen in Table 6, the results of our analysis on the impact of comorbidity measures on improving the fit of the regression model, ECIS had better model discrimination than CCIS. It had a lower AIC and a higher Harrell’s C-statistic (201231 vs. 201400; 0.856 vs. 0.854, respectively).

## 4. Discussion

This population-based cohort study used both individual comorbidities and overall index scores to analyze the discriminative performance of mortality risk in patients with schizophrenia. The Elixhauser Comorbidity Index Score (ECIS) was found to provide better risk-adjustment than CCIS.

This study found 14.1% of our patients to have one or more comorbidities used by the Charlson Comorbidity Index. One population-based study of hospitalized schizophrenia patients in Spain by Bouza et al., that also used the Charlson Comorbidity Index, found that 20% of the patients had a physical comorbidity [23]. Because of changes in the de-institutionalization process instituted in the 1980s, most people with schizophrenia were living in community in Spain at the time of their study [29]. In Taiwan, many patients with schizophrenia, including those without physical illnesses, remain institutionalized in hospitals. Thus, the lower incidence of comorbidities in Taiwan may be related to long-term antipsychotic prescribing in Taiwan which was compatible with the previous Finland study [30].

In the United States, it has been found that patients with schizophrenia have a higher mortality rate than the general population not only due to unnatural deaths such as suicide, homicide, and accidents but also due to natural deaths [31]. In Taiwan, it has been found that these patients are more likely to die from physical illnesses such as myocardial infarction [32], strokes [33], diabetic mellitus [34], renal disease [35], and cancer [36]. Patients with schizophrenia in Taiwan have shorter lives than the general population when they have comorbid physical illnesses [31,37]. Although the Charlson Comorbidity Index was developed in a cohort of 559 medical patients [1] and not a population of patients with schizophrenia, we found that all the items of Charlson comorbidities were statistically associated with the mortality in our schizophrenic population, even after adjustment for demographic data and hospital level. In other words, although the items of the Charlson Comorbidity Index were not developed specifically for patients with schizophrenic disorders, it still has a good discriminative performance on mortality for this population. We found that a diagnosis of any Charlson-listed comorbidity increased the risk of death.

Most of the Elixhauser-listed comorbidities also predicted mortality among our population of patients with schizophrenic disorders. A diagnosis of certain cardiovascular risk factors such as myocardial infarction, congestive heart failure, and peripheral vascular disease in CCIS and congestive heart failure, cardiac arrhythmia, valvular disease, and hypertension but not peripheral vascular disorders in ECIS were associated with a significant increase in the risk of mortality. This finding is consistent with one previous study reporting that adults who have schizophrenia have a greater risk of mortality from cardiovascular and respiratory diseases with modifiable cardiovascular risk factors, including tobacco use, compared to those who do not have schizophrenia [31]. Metastatic cancer has a poor prognosis in all populations and put our patients at great risk of mortality, as other studies have shown [1,38,39]. A probable explanation for this was late or missed diagnosis of the primary cancer. One study in Canada found paralysis, lymphoma, and heart, liver, and renal failure to contribute importantly to increased mortality [39]. With the exception of lymphoma, our findings were similar. With regard to comorbid mental illness, we found six percent of our subjects with schizophrenic disorders had comorbid depression, 14.8% of them dying after hospital discharge. This mortality rate was lower than the total mortality rate (16.6%). Depression is more prevalent in patients with schizophrenia than the general population [40]. In the current study, depression was the most common in all patients but not in those who died. Nevertheless, it remained a predictive factor of mortality in our multivariate analysis. Some physical disorders were reported to be associated with increased suicidal ideation and played a role in the relationship between depression and suicidal ideation among primary care patients [41].

One study of patients admitted to the hospital for any reason found associations between six ECIS-listed comorbid conditions (valvular disease, blood loss anemia, deficiency anemia, obesity, drug abuse, and depression) and decreased mortality [36]. Other studies, too, have reported associations between several morbidities (valvular disease, blood loss, anemia, obesity, depression, hypertension, complicated diabetes, and drug abuse) and a decreased risk of hospital mortality [8,42]. However, our study found no association between any CCIS- or ECIS-listed comorbidities and a reduced risk of mortality. Our study also did not find significant associations between seven ECIS-listed comorbidities (peripheral vascular disorders, hypothyroidism, peptic ulcer disease, lymphoma, rheumatoid arthritis/collagen vascular diseases, obesity, and blood loss anemia) and mortality. There was not enough statistical power with such low numbers of people with peripheral vascular disorders (n = 68) and lymphoma (n = 11). We also found no statistical significance in the association between obesity and mortality. The prevalence of obesity in our study was 0.1%, which was much lower than a previous study of patients with schizophrenia in Taiwan [43]. The difference may be related to study populations. Our data was obtained from the NHIRD. Most doctors in Taiwan did not record obesity diagnosis on insurance claims forms because there was no extra pay for obesity diagnosis. However, obesity was correlated with several diseases including metabolic syndrome, cardiovascular disease which may impact the CCIS or ECIS, and then mortality [44]

Our study showed that, regardless of whether the CCIS or ECIS was used, comorbidity index scores can be used to evaluate mortality risk in patients with schizophrenic disorders after adjustment. Although several ECIS-listed comorbidities did not significantly predict mortality, this index outperformed CCIS in its prediction of mortality.

The strengths of this study included its population-based database, its access to both outpatient and inpatient data and its collection of 17 years of longitudinal data. In addition, our subjects, patients with schizophrenic disorders, were of all ages and died of any cause. However, this study has some limitations. One limitation is that AIDS/HIV is not recorded in NHRID, so we could not evaluate its effect on mortality. Second, the diagnosis of comorbid conditions was based on ICD-9 CM codes. Another limitation is that there is a possibility of misdiagnosis when using claims databases. However, the National Health Insurance Bureau of Taiwan randomly reviews the charts of 1 per 100 ambulatory patients and 1 per 20 inpatients and interviews them to verify the accuracy of the diagnosis and have found it to be reliable. Another limitation is that the severity of comorbidities was not assessed in our study. We identified patients who were newly discharged with schizophrenic disorders in this study, so we did not include those who had never been admitted. The development of CCIS and ECIS were based on one-year all-cause mortality in medical patients. The mortality rate of hospitalized psychiatric patients was lower than medical patients. Therefore, we used 17-year-long follow-up time by CCIS or ECIS to evaluate the mortality risk. We only calculated the ECIS and CCIS during the baseline period and assumed it to be fixed without changes over time. However, a patient will be older and accompanied with more diseases over time, especially in a 17-year-long follow up time. The time-varying issue could attrite the effects of comorbid diseases in mortality risk and introduce the bias. We didn’t separate men from women in calculating CCIS or ECIS, and there was a gender bias toward schizoaffective disorders. Finally, our definition of death was suspension from the National Health Insurance and there was no recording of unnatural causes of death such as accidents or suicide. 

## 5. Conclusions

We concluded that the Elixhauser comorbidity index has better discriminative performance than the Charlson method in its mortality risk of patients with schizophrenic disorders. Psychiatrists should not only treat the mental illness but also physical illness in patients with schizophrenic disorders to reduce mortality. Furthermore, physicians are encouraged to promote the health of patients with schizophrenic disorders by suggesting more exercise and early detection of physical illnesses.

## Figures and Tables

**Table 1 ijerph-17-02450-t001:** Demographic Characteristics and the Hazard Ratio of Mortality.

Characteristics	Total Patients n = 58,771		Mortality n = 9728	
	N (Mean)	% (SD)	HR	95%CI
Age	(37.05)	(12.44)	1.05	1.05–1.05 ***
Gender				
Female	26,142	45.5	1	
Male	32,629	55.5	1.20	1.15–1.25 ***
Geographic region				
Eastern	2713	4.6	1	
Southern	15,672	28.2	0.54	0.50–0.59 ***
Central	13,807	23.5	0.49	0.45–0.53 ***
Northern	25,679	43.7	0.48	0.45–0.52 ***
Socioeconomic status (Enrollee category)				
Low	28,810	49.0	1	
Moderate	13,769	23.4	0.88	0.84–0.93 ***
High	16,192	27.6	0.67	0.63–0.70 ***
Teaching level				
District	15,161	25.8	1	
Regional	30,479	51.9	0.74	0.71–0.78 ***
Medical center	13,131	22.3	0.61	0.58–0.65 ***
Elixhauser Comorbidity Index Score	(0.35)	(2.72)	1.09	1.09–1.10 ***
Charlson Comorbidity Index Score	(0.21)	(0.64)	1.51	1.49–1.54 ***

*** *p* < 0.001; SD = standard deviation, HR = hazard ration, CI = confidence interval.

**Table 2 ijerph-17-02450-t002:** Distribution of Charlson Comorbidities and Mortality in Patient Cohort (n = 58,771).

Charlson Comorbidities	Comorbidity		Mortality	
	n	%	n	%
Myocardial infarction	48	0.1	31	64.6
Congestive heart failure	187	0.3	117	62.6
Peripheral vascular disease	76	0.1	33	43.4
Cerebrovascular disease	832	1.4	368	44.2
Dementia	540	0.9	284	52.6
Chronic pulmonary disease	1865	3.2	574	30.8
Rheumatic disease	773	1.3	205	26.5
Peptic ulcer disease	2443	4.2	616	25.2
Mild liver disease	1379	2.3	407	29.5
Diabetes mellitus without end-organ damage	1956	3.3	744	38.0
Diabetes mellitus with end-organ damage	228	0.4	115	50.4
Hemiplegia	84	0.1	36	42.9
Renal disease	339	0.6	157	46.3
Any malignancy, including lymphoma and leukemia, except malignant neoplasm of skin	277	0.5	129	46.6
Moderate liver disease	75	0.1	54	72.0
Metastatic solid tumor	31	0.1	23	74.2
HIV/AIDS	-	-	-	-

AIDS = acquired immunodeficiency syndrome, HIV = human immunodeficiency virus.

**Table 3 ijerph-17-02450-t003:** Distribution of Elixhauser Comorbidities in Patient Cohort (n = 58,771).

Elixhauser Comorbidities	Comorbidity		Mortality	
	n	%	n	%
Congestive heart failure	236	0.4	150	63.6
Cardiac arrhythmias	446	0.8	137	30.7
Valvular disease	310	0.5	79	25.5
Pulmonary circulation disorders	20	0	14	70.0
Peripheral vascular disorders	68	0.1	23	33.8
Hypertension	2896	4.9	1011	34.9
Paralysis	158	0.3	57	36.1
Neurodegenerative disorders	1201	2.0	329	27.4
Chronic pulmonary disease	1404	2.4	452	32.2
Diabetes, uncomplicated	1714	2.9	628	36.6
Diabetes, complicated	718	1.2	322	44.8
Hypothyroidism	138	0.2	26	18.8
Renal failure	119	0.2	75	63.0
Liver disease	1514	2.6	452	29.9
Peptic ulcer disease excluding bleeding	258	0.4	62	24.0
AIDS/HIV	-	-	-	-
Lymphoma	11	<0.1	3	27.3
Metastatic cancer	30	0.1	23	76.7
Solid tumor without metastasis	245	0.4	118	48.2
Rheumatoid arthritis/collagen vascular diseases	242	0.4	47	19.4
Coagulopathy	53	0.1	20	37.7
Obesity	77	0.1	12	15.6
Weight loss	289	0.5	87	30.1
Fluid and electrolyte disorders	671	1.1	250	37.3
Blood loss anemia	61	0.1	14	23.0
Deficiency anemia	979	1.7	281	28.7
Alcohol abuse	1359	2.3	415	30.5
Drug abuse	975	1.7	180	18.5
Psychoses	58,771	100	9728	16.6
Depression	3520	6.0	522	14.8

AIDS = acquired immunodeficiency syndrome, HIV = human immunodeficiency virus.

**Table 4 ijerph-17-02450-t004:** Adjusted Hazard Rations of Mortality among Patients with Schizophrenic Disorders Based on the Charlson Comorbidities.

Charlson Comorbidities	Univariate Analysis		Multivariate Analysis ^#^	
	HR	95%CI	aHR	95%CI
Myocardial infarction	6.83	4.80–9.72 ***	2.36	1.66–3.36 ***
Congestive heart failure	7.04	5.86–8.45 ***	2.74	2.27–3.29 ***
Peripheral vascular disease	3.52	2.50–4.95 ***	2.05	1.46–2.89 ***
Cerebrovascular disease	3.53	3.18–3.91 ***	1.85	1.67–2.06 ***
Dementia	4.39	3.90–4.95 ***	1.78	1.58–2.01 ***
Chronic pulmonary disease	2.16	1.99–2.35 ***	1.49	1.37–1.62 ***
Rheumatic disease	1.47	1.28–1.69 ***	1.21	1.05–1.39 ***
Peptic ulcer disease	1.68	1.55–1.82 ***	1.36	1.25–1.47 ***
Mild liver disease	2.05	1.86–2.27 ***	1.81	1.64–2.00 ***
Diabetes mellitus without end-organ damage	3.14	2.91–3.39 ***	1.86	1.73–2.01 ***
Diabetes mellitus with end-organ damage	4.66	3.88–5.60 ***	2.53	2.10–3.04 ***
Hemiplegia	3.33	2.40–4.62 ***	2.17	1.56–3.01 ***
Renal disease	3.75	3.20–4.39 ***	2.53	2.15–2.96 ***
Any malignancy, including lymphoma and leukemia, except malignant neoplasm of skin	3.97	3.34–4.73 ***	3.13	2.63–3.72 ***
Moderate liver disease	7.82	5.98–10.21 ***	5.59	4.27–7.31 ***
Metastatic solid tumor	10.08	6.69–15.17 ***	6.31	4.19–9.50 ***
HIV/AIDS	-	-	-	-

*** *p* < 0.001; CI = confidence interval, HR = hazard ratio, aHR = adjusted hazard ratio, AIDS = acquired immunodeficiency syndrome, HIV = human immunodeficiency virus. ^#^ Adjusted for the patient’s age, gender, socioeconomic status, geographic regions, and teaching level of hospital.

**Table 5 ijerph-17-02450-t005:** Adjusted Hazard Ratios of Mortality Among Patients with Schizophrenic Disorders Based on the Elixhauser Comorbidities.

Elixhauser Comorbidities	Univariate Analysis		Multivariate Analysis ^#^	
	HR	95%CI	aHR	95%CI
Congestive heart failure	7.38	6.28–8.67 ***	2.82	2.40–3.32 ***
Cardiac arrhythmias	2.40	2.03–2.84 ***	1.75	1.48–2.07 ***
Valvular disease	1.86	1.49–2.31 ***	1.85	1.48–2.30 ***
Pulmonary circulation disorders	8.09	4.79–13.67 ***	5.66	3.35–9.56 ***
Peripheral vascular disorders	2.75	1.82–4.13 ***	1.19	0.78–1.79
Hypertension	3.02	2.83–3.23 ***	1.48	1.38–1.59 ***
Paralysis	2.70	2.08–3.51 ***	2.12	1.63–2.75 ***
Neurodegenerative disorders	1.98	1.77–2.21 ***	1.56	1.40–1.74 ***
Chronic pulmonary disease	2.53	2.30–2.78 ***	1.59	1.44–1.75 ***
Diabetes, uncomplicated	3.11	2.87–3.38 ***	1.85	1.70–2.01 ***
Diabetes, complicated	3.77	3.37–4.21 ***	2.18	1.95–2.44 ***
Hypothyroidism	1.23	0.84–1.81	1.08	0.73–1.58
Renal failure	7.13	5.68–8.94 ***	3.89	3.10–4.88 ***
Liver disease	2.26	2.05–2.48 ***	1.88	1.71–2.07 ***
Peptic ulcer disease excluding bleeding	1.76	1.37–2.26 ***	1.24	0.96–1.59
AIDS/HIV	-	-	-	-
Lymphoma	1.64	0.53–5.08	1.21	0.39–3.75
Metastatic cancer	11.33	7.52–17.06 ***	6.49	4.31–9.78 ***
Solid tumor without metastasis	4.39	3.66–5.26 ***	3.29	2.74–3.94 ***
Rheumatoid arthritis/collagen vascular diseases	1.35	1.02–1.80 *	1.10	0.83–1.47
Coagulopathy	2.92	1.89–4.53 ***	2.65	1.71–4.11 ***
Obesity	1.21	0.69–2.14	1.44	0.82–2.54
Weight loss	2.17	1.76–2.68 ***	1.63	1.32–2.01 ***
Fluid and electrolyte disorders	3.14	2.77–3.56 ***	2.04	1.80–2.31 ***
Blood loss anemia	1.65	0.98–2.79	1.51	0.89–2.55
Deficiency anemia	2.15	1.91–2.42 ***	1.66	1.47–1.87 ***
Alcohol abuse	2.23	2.02–2.46 ***	2.05	1.86–2.27 ***
Drug abuse	1.15	0.99–1.33	1.67	1.44–1.93 ***
Psychoses	-	-	-	-
Depression	1.06	0.97x1.15	1.17	1.07–1.28 **

* *p* < 0.05, ** *p* < 0.01, *** *p* < 0.001; CI = confidence interval, HR = hazard ratio, aHR = adjusted hazard ratio, AIDS = acquired immunodeficiency syndrome, HIV = human immunodeficiency virus. ^#^ Adjusted for the patient’s age, gender, socioeconomic status, geographic regions, and teaching level of hospital.

**Table 6 ijerph-17-02450-t006:** The 17-year Survival Comparison of Charlson and Elixhauser Comorbidities in Patients with Schizophrenia.

Variable	AIC	Harrell’s c-Statistics
Base model *	202118	0.846
Base model + CCIS (items)	201400	0.854
Base model + Elixhauser (items)	201231	0.856

AIC = Akaike information criterion. * Base model included age, gender, socioeconomic status, geographic area, and teaching level of hospital.

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
