# Peer review of "Comparison of Elixhauser and Charlson Methods for Discriminative Performance in Mortality Risk in Patients with Schizophrenic Disorders"

_ijerph, 2020, doi:10.3390/ijerph17072450_

Round 1

Reviewer 1 Report

An interesting study that adds to the international literature and evidence regarding important differences between in-patient and community populations of people with schizophrenia disorders, and their risk of mortality. I refer to page and line numbers in the PDF.

It is not yet clear in the methodology section what version of the Charleston Comorbidity Index was used in your study to calculate the CCIS. I mention this because there are different versions (mapped to ICD-09, and later ICD-10) and we know that the NHIRD data in the time frame selected in your study used ICD-09-CM codes (Page 2, line 74).

That 60% excess mortality in schizophrenia is due to physical illness is a sweeping statement on Page 2, line 55. You have attributed this to one citation that is not a primary source. I would prefer to see a range, or state that up to 60% may be due to physical illness. We know, for example, (from meta analyses that have calculated SMR estimates) that mortality in schizophrenia is 2-3 times that of the general population, with the highest rates of mortality reported in men with schizophrenia and women with a diagnosis of schizoaffective disorder.  Suicide is significant in the early years after diagnosis, cardiovascular disease seems to have the largest impact after this (in community populations).

ICD-09-CM code 295 is the code for "schizophrenic disorders" and I recommend that this grouping of disorders in the study population is acknowledged (Page 3, line 87 onwards). This is because this class includes many different types of schizophrenia diagnosis, including schizoaffective disorder (that has a slightly different comorbidity profile than schizophrenia, particularly in women).

Although your study appears to use exactly the same method of comparison and statistical analyses as the paper published by Chang et al in 2016 that  compared the risk of mortality in patients with oral cancers you report that you adjusted differences between the two comorbidity indices in the same way as these authors (Page 3, 108-109). However I cannot identify details of this adjustment in the original paper (only that there were slightly different comorbidity groupings for similar disorders between the two indices). I therefore recommend you include some more information about why and how adjustments were made in your study, and that you also disclose and aknowledge that the method used in the design of your study was taken directly from this first study that was published by different authors. 

It is not yet clear where the record of death was taken from (Page 2, 113). I assume the NHIRD but if so, please make this explicit.

There is a missing citation for the Akaike Information Criterion [AIC] statistical test and it is not clear if all assumptions were met for its appropriate use. For example did you consider using a corrected version (AICc) due to your sample size and, if so, why did you decide not to use it?

 It is an important observation that the lower incidence of comorbidity in your sample may be due to the inpatient population (“closer medical care” Page 9, 207). I would consider comparing this with studies in Scandinavian countries that have had similar findings when comparing datasets from before and after the introduction of community care or that consider long term antipsychotic prescribing as a proxy for engagement with "closer" care (e.g. Tiihonen et al, 2009).

On Page 9, 229 Although you state you and others have found that metastatic cancer predicts mortality there is a need to acknowledge that this diagnosis has a poor prognosis in all populations. The issue in schizophrenia is probably more due to late or missed diagnosis of the primary cancer. Similarly, when discussing the risk of depression (also page 9) I think it is important to acknowledge if this is linked to all mortality, or a feature of suicide or both. We know that depression leads to worse outcomes in most long-term physical comorbidities (e.g. heart disease).

Page 9, 248. This sentence does not make sense. I think it is pretty clear that there was not enough statistical power with such low numbers of people with lymphoma.

You make an interesting point about the diagnosis of obesity not commonly being recorded by Doctors in Taiwan. I wonder if there were any other cultural, professional or organisational issues like this that  also could also have impacted on the results. For example, the reporting and recording of unnatural causes of death (accidents, suicide),  or a gender bias towards affective (and therefore schizoaffective disorder) diagnosis in women.

There are a few grammatical errors that can be addressed through careful proof reading. For example on Page 2, line 71 "the data provided the Taiwan’s Bureau" should be "the data provided from the Taiwan’s Bureau".

Author Response

Dear Editor:

Manuscript ID: IJERPH_746792 “Comparison of Elixhauser and Charlson Methods for Predicting Schizophrenia Patients Mortality”

We are grateful for the valuable comments from the editors and reviewers on our manuscript. We would like to thank the reviewers for the considering our manuscript interesting and gave us many information. The following responses have been prepared to address all of the reviewers’ comments in a point-by-point fashion. We also revised our manuscript following all of the reviewers’ suggestions.

Please let us know anything else we should provide.

Sincerely yours

Yu-Mei Chou

Department of Anesthesiology, Kaohsiung Veterans General Hospital

No.386, Dazhong 1st Rd., Kaohsiung, Taiwan

E-mail: ymchou@vghks.gov.tw

For Reviewer 1:

An interesting study that adds to the international literature and evidence regarding important differences between in-patient and community populations of people with schizophrenia disorders, and their risk of mortality. I refer to page and line numbers in the PDF.

Comment

  1. It is not yet clear in the methodology section what version of the Charleston Comorbidity Index was used in your study to calculate the CCIS. I mention this because there are different versions (mapped to ICD-09, and later ICD-10) and we know that the NHIRD data in the time frame selected in your study used ICD-09-CM codes (Page 2, line 74).

Response

Thank you for your comment.

We modified the comorbidity identification section as following, “We used diagnoses codes (ICD-9-CM codes) from administrative and ambulatory claims record to calculate the Charlson and Elixhauser index by using Cheng et al’s (2016) version [9].” Please refer to line 119-121.

Comment

  1. That 60% excess mortality in schizophrenia is due to physical illness is a sweeping statement on Page 2, line 55. You have attributed this to one citation that is not a primary source. I would prefer to see a range, or state that up to 60% may be due to physical illness. We know, for example, (from meta analyses that have calculated SMR estimates) that mortality in schizophrenia is 2-3 times that of the general population, with the highest rates of mortality reported in men with schizophrenia and women with a diagnosis of schizoaffective disorder. Suicide is significant in the early years after diagnosis, cardiovascular disease seems to have the largest impact after this (in community populations).

Response

Thank you for your comment.

    We addressed the mortality in patients with schizophrenia group in the Introduction section as following, “In a meta-analysis, increased overall mortality rates were noted in patients with schizophrenia and other nonaffective psychotic disorders (standardized mortality ratio (SMR)= 3.09, 95% CI 2.9–3.3) [20]. Increased mortality rates for suicide was 12 to more than 20 times expected and increased standardized mortality ratio from natural causes was at least 2 times expected in patients with schizophrenia [21].” Please refer to line 56-60.

Comment

  1. ICD-09-CM code 295 is the code for "schizophrenic disorders" and I recommend that this grouping of disorders in the study population is acknowledged (Page 3, line 87 onwards). This is because this class includes many different types of schizophrenia diagnosis, including schizoaffective disorder (that has a slightly different comorbidity profile than schizophrenia, particularly in women).

Response

Thank you for your comment.

We modified the study population as following, “Patients with schizophrenic disorders, including schizophrenia and schizoaffective disorders were identified by the ICD-9-CM code 295 listed on their claims submissions.” Please refer to line 100-101.

Comment

  1. Although your study appears to use exactly the same method of comparison and statistical analyses as the paper published by Chang et al in 2016 that compared the risk of mortality in patients with oral cancers you report that you adjusted differences between the two comorbidity indices in the same way as these authors (Page 3, 108-109). However I cannot identify details of this adjustment in the original paper (only that there were slightly different comorbidity groupings for similar disorders between the two indices). I therefore recommend you include some more information about why and how adjustments were made in your study, and that you also disclose and aknowledge that the method used in the design of your study was taken directly from this first study that was published by different authors.

Response

Thank you for your comment.

We modified the comorbidity identification section as following, “We used diagnoses codes (ICD-9-CM codes) from administrative and ambulatory claims record to calculate the Charlson and Elixhauser index by using Cheng et al’s (2016) version [9]. ” Please refer to line 119-120.

Comment

  1. It is not yet clear where the record of death was taken from (Page 2, 113). I assume the NHIRD but if so, please make this explicit.

Response

Thank you for your comments. We stated the data sources in Study design and data sources section as following “All data, including schizophrenic disorders, numerous comorbidity conditions and death, were collected from National Health Insurance Research Database (NHIRD).” Please refer to line 76-77.

Comment

  1. There is a missing citation for the Akaike Information Criterion [AIC] statistical test and it is not clear if all assumptions were met for its appropriate use. For example did you consider using a corrected version (AICc) due to your sample size and, if so, why did you decide not to use it?

Response

Thank you for your comments.

We added the reference of Akaike Information Criterion [AIC] as following, “H. Akaike, "A new look at the statistical model identification," in IEEE Transactions on Automatic Control, vol. 19, no. 6, pp. 716-723, December 1974.”.

The authors agree the importance of AICc and have considered AICc instead of AIC for analysis. However, when the number of observations is large, the Akaike Information Criterion (AIC) and the small-sample corrected Akaike Information Criterion (AICc) become extremely similar because AICc converges to AIC. Therefore, we used AIC in this study.

Comment

  1. It is an important observation that the lower incidence of comorbidity in your sample may be due to the inpatient population (“closer medical care” Page 9, 207). I would consider comparing this with studies in Scandinavian countries that have had similar findings when comparing datasets from before and after the introduction of community care or that consider long term antipsychotic prescribing as a proxy for engagement with "closer" care (e.g. Tiihonen et al, 2009).

Response

Thank you for your comments.

We addressed the “closer care” issue in the Discussion section as following, “Thus, the lower incidence of comorbidities in Taiwan may be related to long term antipsychotic prescribing in Taiwan which was compatible with the previous Finland study [30]. “ Please refer to line 222-223.

Comment

  1. On Page 9, 229 Although you state you and others have found that metastatic cancer predicts mortality there is a need to acknowledge that this diagnosis has a poor prognosis in all populations. The issue in schizophrenia is probably more due to late or missed diagnosis of the primary cancer. Similarly, when discussing the risk of depression (also page 9) I think it is important to acknowledge if this is linked to all mortality, or a feature of suicide or both. We know that depression leads to worse outcomes in most long-term physical comorbidities (e.g. heart disease).

Response

Thank you for your comments.

We addressed the “metastatic cancer” issue in the discussion section as following “Metastatic Cancer has a poor prognosis to all population and put our patients at great risk of mortality, as have other studies [1,35,36]. A probable explanation was late or missed diagnosis of the primary cancer.” Please refer to line 245-247.

We addressed the “depression” issue in the discussion section as following “Some physical disorders were reported to be associated with increased suicidal ideation and play a role in the relationship between depression and suicidal ideation among primary care patients [41].” Please refer to line 254-256.

Comment

  1. Page 9, 248. This sentence does not make sense. I think it is pretty clear that there was not enough statistical power with such low numbers of people with lymphoma.

You make an interesting point about the diagnosis of obesity not commonly being recorded by Doctors in Taiwan. I wonder if there were any other cultural, professional or organisational issues like this that  also could also have impacted on the results. For example, the reporting and recording of unnatural causes of death (accidents, suicide),  or a gender bias towards affective (and therefore schizoaffective disorder) diagnosis in women.

Response

Thank you for your comments

We’ve modified the lymphoma issue in the discussion section as following “ There was not enough statistical power with such low numbers of people with peripheral vascular disorders (n=68) and lymphoma (n=11)” . Please refer to line 265-268.

We’ve modified the obesity issue in the discussion section as following “ Our data was obtained from the NHIRD. Most doctors in Taiwan do not record obesity diagnosis on insurance claims forms because there was no extra pay for obesity diagnosis. However, obesity was correlated with several diseases including metabolic syndrome, cardiovascular disease which may impact the CCIS or ECIS and then mortality [44].” Please refer to line 272-274.

We’ve modified the gender bias issue in the discussion section as following “ We didn’t separate men from women in calculating CCIS or ECIS, and there was a gender bias toward schizoaffective disorders.” Please refer to line 293-294.

We’ve modified the unnatural death bias issue in the discussion section as following “ Our definition of death was suspension from the National Health Insurance and there was no recording of unnatural causes of death such as accidents or suicide.” Please refer to line 294-296.

Comment

  1. There are a few grammatical errors that can be addressed through careful proof reading. For example on Page 2, line 71 "the data provided the Taiwan’s Bureau" should be "the data provided from the Taiwan’s Bureau".

Response

Thank you for your comments. We’ve examined the manuscript and corrected typos.

Reviewer 2 Report

This manuscript aims to evaluate the predicting accuracy between Elixhauser (ECIS) and Charlson comorbidity index (CCIS) in patients with schizophrenia. The data is well illustrated and conclusions stay with the results. However, the manuscript must be much improved before publication:

  • Although the results provide AIC scores to evaluate the levels of goodness of fit among different models, they only can provide the “internal validity” between data and models. If authors want to prove the predicting accuracy, the “testing data” (e.g., the NHIRD in other years, not during study period) are needed to provide the “external validity” between different models. If ECIS is still has a lower AIC scores and higher Harrells C-statistic by using “testing data”, it will prove that ECIS in predicting accuracy is better than CCIS. Without the results of testing data, the authors should not claim that ECIS outperforms than CCIS.
  • 17-year overall mortality is a much long follow-up period. According to this manuscript, the ECIS and CCIS were only calculated during baseline period (the start of follow-up time) and a patient’s ECIS and CCIS was considered as fixed without changes over time. Unfortunately, it is impossible during such long follow-up time in real world. A patients will be older and accompanied with more diseases over time. Therefore, the scores of ECIS and CCIS will higher with long follow-up time. The time-varying issue could attrite the effects of comorbid diseases in mortality risk and introduce the bias.

Author Response

Dear Editor:

Manuscript ID: IJERPH_746792 “Comparison of Elixhauser and Charlson Methods for Predicting Schizophrenia Patients Mortality”

We are grateful for the valuable comments from the editors and reviewers on our manuscript. We would like to thank the reviewers for the considering our manuscript interesting and gave us many information. The following responses have been prepared to address all of the reviewers’ comments in a point-by-point fashion. We also revised our manuscript following all of the reviewers’ suggestions.

Please let us know anything else we should provide.

Sincerely yours

Yu-Mei Chou

Department of Anesthesiology, Kaohsiung Veterans General Hospital

No.386, Dazhong 1st Rd., Kaohsiung, Taiwan

E-mail: ymchou@vghks.gov.tw

For Reviewer 2:

This manuscript aims to evaluate the predicting accuracy between Elixhauser (ECIS) and Charlson comorbidity index (CCIS) in patients with schizophrenia. The data is well illustrated and conclusions stay with the results. However, the manuscript must be much improved before publication:

Comment

  1. Although the results provide AIC scores to evaluate the levels of goodness of fit among different models, they only can provide the “internal validity” between data and models. If authors want to prove the predicting accuracy, the “testing data” (e.g., the NHIRD in other years, not during study period) are needed to provide the “external validity” between different models. If ECIS is still has a lower AIC scores and higher Harrells C-statistic by using “testing data”, it will prove that ECIS in predicting accuracy is better than CCIS. Without the results of testing data, the authors should not claim that ECIS outperforms than CCIS.

Response

Thank you for your comment.

We modified the abstract section as following, “ECIS had better discriminative performance in mortality risk than CCIS in patients with schizophrenic disorders.” Please refer to line 29-31.

We modified the abstract and conclusion section as following, “We concluded that Elixhauser comorbidity index has better discriminative performance than the Charlson method in its mortality risk of patients with schizophrenic disorders.” Please refer to line 298-300.

Comment

  1. 17-year overall mortality is a much long follow-up period. According to this manuscript, the ECIS and CCIS were only calculated during baseline period (the start of follow-up time) and a patient’s ECIS and CCIS was considered as fixed without changes over time. Unfortunately, it is impossible during such long follow-up time in real world. A patients will be older and accompanied with more diseases over time. Therefore, the scores of ECIS and CCIS will higher with long follow-up time. The time-varying issue could attrite the effects of comorbid diseases in mortality risk and introduce the bias.

Response

Thank you for your comment.

    We addressed the long follow-up time in discussion section as following, “We only calculated the ECIS and CCIS during baseline period and assumed fixed without changes over time. However, a patient will be older and accompanied with more disease over time, especially in 17-year-long follow up time. The time-varying issue could attrite the effects of comorbid diseases in mortality risk and introduce the bias.” Please refer to line 290-293.

Round 2

Reviewer 2 Report

The manuscript in methods had been improved by authors. In order to make this manuscript completeness, there is an important result should be provided before publication:

  1. The development of CCIS and ECIS scores were based on one-year all-cause mortality. Therefore, the authors should also provide the discriminative performance in one-year all-cause mortality and discuss the difference in different population and length of follow-up time. This results will make this manuscript more completeness and more scientific significance.

Author Response

Dear Editor:

Manuscript ID: IJERPH_746792 “Comparison of Elixhauser and Charlson Methods for Predicting Schizophrenia Patients Mortality”

We are grateful for the valuable comments from the editors and reviewers on our manuscript. We would like to thank the reviewers for the considering our manuscript interesting and gave us many information. The following responses have been prepared to address all of the reviewers’ comments in a point-by-point fashion. We also revised our manuscript following all of the reviewers’ suggestions.

Please let us know anything else we should provide.

Sincerely yours

Yu-Mei Chou

Department of Anesthesiology, Kaohsiung Veterans General Hospital

No.386, Dazhong 1st Rd., Kaohsiung, Taiwan

E-mail: ymchou@vghks.gov.tw

For Reviewer 2:

The manuscript in methods had been improved by authors. In order to make this manuscript completeness, there is an important result should be provided before publication:

Comment

  1. The development of CCIS and ECIS scores were based on one-year all-cause mortality. Therefore, the authors should also provide the discriminative performance in one-year all-cause mortality and discuss the difference in different population and length of follow-up time. This results will make this manuscript more completeness and more scientific significance.

Response

Thank you for your comment.

We modified the limitation section as following, “The development of CCIS and ECIS were based on one-year all-cause mortality in medical patients. The mortality rate of hospitalized psychiatric patients was lower than medical patients. Therefore, we used 17-year-long follow-up time by CCIS or ECIS to evaluating mortality risk. We only calculated the ECIS and CCIS during baseline period and assumed fixed without changes over time. However, a patient will be older and accompanied with more diseases over time, especially in 17-year-long follow up time. The time-varying issue could attrite the effects of comorbid diseases in mortality risk and introduce the bias.” Please refer to line 284-290.